



# The response of mesospheric $H_2O$ and CO to solar irradiance variability in models and observations

Arseniy Karagodin-Doyennel[1,2,*], Eugene Rozanov[1,2,*], Ales Kuchar[3,*], William Ball[4,*],
Pavle Arsenovic[5], Ellis Remsberg[6], Patrick Jöckel[7], Markus Kunze[8], David A. Plummer[9],
Andrea Stenke[1], Daniel Marsh[10,11], Doug Kinnison[10], and Thomas Peter[1]

[1]The Institute for Atmospheric and Climate Science (IAC) ETH, Zurich, Switzerland
[2]The Physikalisch-Meteorologisches Observatorium Davos/World Radiation Center (PMOD/WRC), Davos, Switzerland
[3]Leipzig Institute for Meteorology (LIM), Leipzig, Germany
[4]Department of Geoscience and Remote Sensing, TU Delft, Delft, Netherlands
[5]Swiss Federal Laboratories for Materials Science and Technology (EMPA), Dübendorf, Switzerland
[6]Science Directorate NASA Langley Research Center Hampton, Virginia, USA
[7]Institute of Atmospheric Physics, Wessling, Germany
[8]Institut für Meteorologie, Freie Universität Berlin, 12165 Berlin, Germany
[9]Climate Research Division, Environment and Climate Change Canada, Montreal, Canada
[10]National Center for Atmospheric Research, Boulder, Colorado, USA
[11]Priestley International Centre for Climate, University of Leeds, UK
[*]These authors contributed equally to this work.

**Correspondence:** Arseniy Karagodin-Doyennel (darseni@student.ethz.ch)

**Abstract.**

Water vapor ($H_2O$) is the source of reactive hydrogen radicals in the middle atmosphere, whereas carbon monoxide (CO), being formed by $CO_2$ photolysis, is suitable as a dynamical tracer. In the mesosphere, both $H_2O$ and CO are sensitive to solar irradiance variability because of their destruction/production by solar radiation. This enables to analyze the solar signal
in both, models and observed data. Here, we evaluate the mesospheric $H_2O$ and CO response to solar irradiance variability using the Chemistry-Climate Model Initiative (CCMI-1) simulations and satellite data. We analyzed the results of four CCMI models (CMAM, EMAC-L90MA, SOCOLv3, CESM1-WACCM 3.5) operated in CCMI reference simulation REF-C1SD in specified dynamics mode, covering the period from 1984 to 2017. Multiple linear regression analysis shows a pronounced and statistically robust response of $H_2O$ and CO to solar irradiance variability, and to the annual and semiannual cycles. For periods
with available satellite data, we compared the simulated solar signal against satellite observations, namely during 1992-2017 for $H_2O$ and 2005-2017 for CO. The model results generally agree with observations and reproduce an expected negative and positive correlation for $H_2O$ and CO, respectively, with solar irradiance. However, the magnitude of the response and patterns of the solar signal varies among the considered models, indicating differences in the applied chemical reaction and dynamical schemes including the representation of photolyses. We suggest that there is no dominating thermospheric influence of solar
irradiance in CO, as reported in previous studies because the response to solar variability is comparable with observations in both, low-top and high-top models. We stress the importance of this work for improving our understanding of the current ability and limitations of state-of-the-art models to simulate a solar signal in the chemistry and dynamic of the middle atmosphere.





## 1 Introduction

$H_2O$ plays an important role in atmospheric chemistry as a source of the hydrogen oxide radicals ($HO_x$), which are important

for ozone loss. There are two main sources of water vapor in the middle atmosphere. The first is a direct carry-over of $H_2O$ through the tropopause tropical cold trap ($\sim$ 2-3 ppmv), where strong dehydration of air occurs (Nicolet, 1981). The second is indirect, namely the upward stratospheric transport of $CH_4$ and its subsequent oxidation. The main chemical reaction leading to $H_2O$ formation throughout the atmosphere is from methane oxidation (Wofsy et al., 1972):

$$CH_4 + OH^\cdot \rightarrow CH_3^\cdot + H_2O. \tag{1}$$

Middle atmospheric trends in $H_2O$ are largely determined by changes in the tropospheric content of $CH_4$ and temperature at the tropical tropopause (Nedoluha et al., 2013). The amount of $H_2O$ in the middle atmosphere can reach the value up to 10 ppmv (Brasseur and Solomon, 2005). In the mesosphere where the $H_4$ is fully oxidized, the $H_2O$ can have the amount of about 6.6 ppmv. Newerthless, the highest mixing ratio of $H_2O$ is in the lower atmosphere with 10-100 ppm in the upper and more than ten thousand ppm in the lower troposphere (Palchetti et al., 2008). With increasing altitude in the mesosphere,

photodissociation of $H_2O$ is caused by solar irradiance at the Ly-$\alpha$ (121.25 nm) spectral line of hydrogen and within the spectral range of the oxygen Schumann-Runge continuum (175-200 nm; Frederick and Hudson, 1980). The photodissociation lifetime of water vapor in the presence of the solar Ly-$\alpha$ radiation below mesopause is estimated to be less than 200 hours (Kingston, 1987) because $J_{(Ly-\alpha)}$ of ($H_2O$) = 1.6 x $10^{-6}$ $s^{-1}$ for total number of $O_2$ molecules of about $10^{20}$ $cm^{-2}$ . Products of $H_2O$ photolysis are atomic hydrogen and hydroxyl radicals:

$$H_2O + h\nu \rightarrow H^\cdot + OH^\cdot. \tag{2}$$

As such, an anti-correlation of water vapor with solar irradiance, with the strongest response in the mesosphere, is expected (Chandra et al., 1997; Hervig and Siskind, 2006; Shapiro et al., 2012) and the strength of this effect depends upon the intensity of solar irradiance in the Ly-$\alpha$ line and the Schumann-Runge band.

Carbon monoxide (CO) is widely present in the lower thermosphere and mesosphere and due to its chemical lifetime of

more than one month, it can be used for investigating transport processes in the middle atmosphere. CO can react with some species (e.g. $OH^\cdot$), which would otherwise destroy ozone and $CH_4$, enhancing its radiative forcing (Ryan et al., 2018). Contrary to $H_2O$, CO is positively correlated with solar irradiance as it is primarily formed through the photolysis of $CO_2$ in the lower thermosphere and upper mesosphere at Ly-$\alpha$ (Wofsy et al., 1972) as follows:

$$CO_2 + h\nu \rightarrow CO + O. \tag{3}$$

In the troposphere, the main source of CO is the oxidation of hydrocarbons (Minschwaner et al., 2010). However, in the mesosphere the amount of CO from the oxidation of $CH_4$ and isoprene is so much smaller compared to the $CO_2$ photodisso-





ciation (Eq 3) that this process can be neglected at high altitudes (Garcia et al., 2014). Chemical loss of CO in the atmosphere occurs by oxidation (Levy, 1971):

$$CO + OH \cdot \rightarrow CO_2 + H \cdot . \tag{4}$$

The amount of CO in the mesosphere is estimated to be within 30ppb-10 ppm (Brasseur and Solomon, 2005), and 50-100 ppb in the uncontaminated air in the troposphere (Minschwaner et al., 2010), having a strong vertical gradient. Mesospheric concentrations of $H_2O$ and CO are strongly determined by the solar irradiance. Since the processes leading to $H_2O$/CO destruction/production are much faster than changes in solar irradiance on all timescales, we can assume they are essentially linear. Therefore, an attribution approach using multiple linear regression (MLR) analysis is reasonable to estimate the impact of so-

lar irradiance on $H_2O$ and CO variability in the middle atmosphere. We apply this linear statistical tool to different model and satellite data. One major goal of this study is to compare the modeled solar signal in mesospheric $H_2O$ and CO to observations. Recently, the photochemical $H_2O$ loss by Ly-$\alpha$ radiation in UARS/HALOE measurements was estimated to be about 35% at 0.01 hPa ($\sim$ 80 km altitude) at 50°N using MLR (Remsberg et al., 2018). Tropical tendencies in mesospheric water vapor using MLR analysis of Aura/MLS observations for the 2004-2015 period were presented by Nath et al. (2018). Their analysis

showed a pronounced trend in water vapor throughout the whole considered period, as well as a strong negative correlation with the F10.7 solar index that maximizes at 0.01 hPa (-0.56 ppmv/ 1% of Ly-$\alpha$). A solar signal in lower stratospheric $H_2O$ was investigated by Schieferdecker et al. (2015). Using MLR they showed a negative correlation between $H_2O$ and solar activity with a phase-shift of about 2 years in composite data of HALOE and MIPAS over 60°N-60°S.

   Lee et al. (2013) presented a study of the middle atmospheric CO variation caused by solar irradiance changes using MLS and

solar irradiance measurements from the Solar Radiation and Climate Experiment (SORCE). Their results reveal a significant positive correlation of up to 0.6 between solar irradiance and CO variation in the mesosphere, as well as downward transport of the CO anomaly induced by solar irradiance over high latitudes with a descent rate of about 1.3 km/day. Lee et al. (2018) expanded their previous work and investigated the solar cycle variation in CO using MLS measurements for 2004-2017, as well as free-running WACCM simulations using two different solar spectral irradiance datasets. The updated results have a

higher correlation (up to 0.8) and show that 68% of upper mesospheric CO variation is caused by solar irradiance changes as well as pronounced downwelling of the signal within the polar vortex regions. The results simulated with WACCM (3.5) underestimate the CO variation in the upper mesosphere by a factor of three compared to the Aura/MLS observations. However, here it should be mentioned that the applied WACCM version does not employ the extreme ultraviolet (EUV) photolysis and reaction by $CO_2$ with $O^+$ as an additional CO production mechanism in the thermosphere. The modeled CO distribution with

the WACCM version 4.0 shows CO in better agreement with the MIPAS and ACE-FTS observations (Garcia et al., 2014). This also could cause some issues when comparing the results of models where this production mechanism is not included. Thus, the results of previous studies revealed issues in the modeling of the influence of solar irradiance and motivates for an inter-comparison analysis of multiple models and observations. So far, an MLR analysis using multiple chemistry-climate models (CCMs) and observations of both CO and $H_2O$ had not been conducted.





In this work, we present an MLR analysis of simulations with several chemistry-climate models in specified dynamics mode for the period 1984-2017, as well as available observations from UARS/HALOE (1992-2005) and Aura/MLS (2005-2017), which provide data for 26 years with a good resolution and without serious gaps. The combined UARS/HALOE and Aura/MLS records provide observations of CO (only for the Aura/MLS period) and $H_2O$ (for the whole 1992-2017 period), which makes these data suited for our analysis. The MLR method is used to retrieve $H_2O$ and CO responses to solar irradiance

variability, and to estimate the consistency of the solar signal in CCMs with respect to that found in observations, and between CCMs. Analyzing the differences in the solar responses can reveal potential model limitations, such as the dynamics of the middle atmosphere (weak or strong transport), presence of thermospheric sources (important since some models have an upper boundary at 0.01 hPa), and photochemistry and chemical production or loss of the species considered here. $H_2O$ and CO were chosen as they are very sensitive to solar irradiance variations in the mesosphere (Remsberg et al., 2018; Lee et al., 2018)

making them good candidates for this kind of analysis.

    In Section 2, we describe the data sets used in this study. Section 3 briefly describes the MLR model set-up used to retrieve the solar signal response. The results of the MLR analysis of the models CMAM, EMAC-L90MA (hereinafter will be denoted as EMAC), SOCOLv3 (hereinafter will be denoted as SOCOL), and WACCM REF-C1SD runs for the entire period 1984-2017, as well as the comparison with $H_2O$ measurements from UARS/HALOE and Aura/MLS for the 1992-2017 period and

CO measurements from Aura/MLS for the 2005-2017 period, are presented in Section 4. The discussion and overall summary can be found in Sections 5 and 6.

## 2    Data sets

For our study chose four global climate models involved in the Chemistry–Climate Model Initiative (CCMI-1) project. The CCMI project aims at carrying out the inter-model comparison and validation of model results with observations[1]. For the

analysis, we used the results of the REF-C1SD experiment which was performed using boundary conditions extracted from observations including the atmospheric level of greenhouse gases and ozone-depleting substances (ODSs), as well as sea surface temperature and sea ice concentration (Morgenstern et al., 2017). Specified dynamics (SD) here means that meteorological fields in the model experiments are nudged toward reanalysis datasets. The nudging is applied in CCMI-1 models for different atmospheric regions as well as using different reanalysis data (see Table 1; Chrysanthou et al., 2019). The selection of models

was based on the inspection of the simulated $H_2O$ and CO time series for the presence of the solar signal in the mesosphere and on data reliability. Careful analysis of the CCMI-1 results showed that only CMAM, EMAC, SOCOL, and WACCM CCMs are suitable for the intended analysis, while other models involved in CCMI-1 were either in an unusable format, did not extend high enough or lack any solar signal in $H_2O$ and CO. The REF-C1SD simulations of the four chosen models were extended to 2017 (CCMI-1 is until 2011) to overlap with the recent satellite measurements.


---

[1]More information on CCMI activities can be found here: https://www.sparc-climate.org/activities/ccm-initiative





Table 1: CCMI-1 model set up

| Name | Spatial resolution | Model top height | Nudging region | Nudging data | Reference |
|---|---|---|---|---|---|
| CMAM | T47, L71 | 0.0008 hPa | Surface–1 hPa | ERA-Interim | Scinocca et al. (2008) |
| EMAC-L90MA | T42, L90MA | 0.01hPa | 10–90 hPa | ERA-Interim | Jöckel et al. (2010); Jöckel et al. (2016) |
| SOCOLv3 | T42, L39 | 0.01 hPa | Surface–0.01 hPa | ERA-Interim | Stenke et al. (2012); Revell et al. (2015) |
| CESM1-WACCM 3.5 | 1.9 x 2.5, L66 | $5.1 \times 10^{-6}$ hPa | Surface–50 km (fades out 40–50 km) | MERRA | Marsh et al. (2013); Verronen et al. (2016) |

We focus on mesospheric altitudes for the examination of the solar signal response in atmospheric chemistry. Thus, differences in nudging setups play no role, as the mesosphere does not undergo direct nudging. There is an exception for SOCOL that the only one model where the whole model atmosphere is nudged up to the 0.01 hPa level. Additionally, in the frame of this work, it is important to describe the lower limit of the wavelength for photolysis and photoionization in CCMI-1 models presented in Table 1.

In EMAC, for the simulation considered in this work, the photolysis rates have been calculated with the submodel JVAL (Sander et al., 2014), which uses 8 wavelength bands, bands ranging from 178.6 nm to 682.5 nm (Landgraf and Crutzen, 1998) and includes a parametrization for Ly-α photolysis (Chabrillat and Kockarts, 1997). In SOCOL photolysis rates are calculated using a look-up-table approach (Rozanov et al., 1999), including effects of the solar irradiance variability with the lower limit for photolysis at 120 nm. In the CMAM model, the shortest wavelength is 121.0 nm. Also, the parameterization for NO photolysis from Minschwaner and Siskind (1993) is used, however, there is no effect of solar variability included on this rate. In WACCM, the photolysis of $H_2O$ starts at Ly-α (121.5 nm). Fluxes at that wavelength are calculated using the Chabrillat and Kockarts (1998) scheme. For Equation 3, cross-sections from 0.5 to 1050.0 nm in the XUV/X-ray wavelength region are used. Solar fluxes are calculated with Solomon and Qian (2005). Additionally, in WACCM an ion chemistry loss for $CO_2$ is included: $CO_2 + O^+ \rightarrow O_2^+ + CO$. In the other models (EMAC, CMAM and SOCOL) considered here, ion chemistry is not included.

Since time series of $H_2O$ and CO from CMAM, SOCOL, WACCM, and EMAC SD simulations are available until 2017, we compare the solar response with observations from Aura/MLS CO for the available period of 2005-2017 and $H_2O$ for 1992-2017. However, to extend the REF-C1SD simulations of SOCOL and EMAC the NRLSSI data (Lean et al., 2005) for REF-C1 was used only until 2011, and onward the models used the boundary conditions (GHGs and ODSs) of the RCP6.0 scenario (REF-C2). In EMAC the conditions of the year 2011 have been cyclically repeated for the years 2012-2017. In case of solar forcing, EMAC uses the adapted solar forcing according to the one used in HadGEM2-ES CMIP5 6.0 simulation





(Jones et al., 2011). The CMAM data for the considered period was from a different specified dynamics simulation than the one submitted to CCMI-1, produced using a method identical to that of nudging with reanalysis but with specified stratospheric

aerosols, extra-terrestrial solar flux and emissions from datasets specified for CMIP6 (Eyring et al., 2016). For the extension of the WACCM time series of both $H_2O$ and CO, the NRLSSI2 model (Coddington et al., 2016) is used from 2015 onward.

To compare simulated results, the observations of $H_2O$ from the Halogen Occultation Experiment HALOE (1992-2005) onboard of the Upper Atmosphere Research Satellite (UARS), and the observations of $H_2O$ and CO from Microwave Limb Sounder (MLS) (2005-2017) instrument on board of the Aura satellite were analyzed. HALOE measured the reduction in

the intensity of solar energy that passes through the atmosphere to obtain the gas concentration of important atmopsheric trace gases. A detailed HALOE instrument description can be found in Russell et al. (1993). The principal method used with the MLS instrument is the measurement of microwave thermal emissions from the atmosphere to remotely obtain profiles of different atmospheric constituents. More information on MLS can be found in Waters et al. (2006). For the analysis of $H_2O$, we used a combination of the GOZCARDS merged data set consisting of all available data for the 1992-2004 period (Anderson

et al., 2013) and data from ongoing missions of MLS (Waters et al., 2006) and ACE-FTS (Atmospheric Chemistry Experiment - Fourier Transform Spectrometer), (Bernath et al., 2005) for 2005-2017 obtained using an averaging procedure based on overlap periods. Carbon monoxide time series are available only for the period 2005-2017 (Bernath et al., 2005; Waters et al., 2006). Both datasets of observations are binned into 20 latitude zones, as data of observations (especially HALOE) are rather noisy and a linear gap-filling procedure was applied to produce a continuous time series.

Figure 1 shows the time series of $H_2O$ and CO averaged over the tropics ($30^o$N-$30^o$S) at 0.01 hPa from CCMI-1 REF-C1SD simulations and observations from the GOZCARDS composite and Aura/MLS instruments.





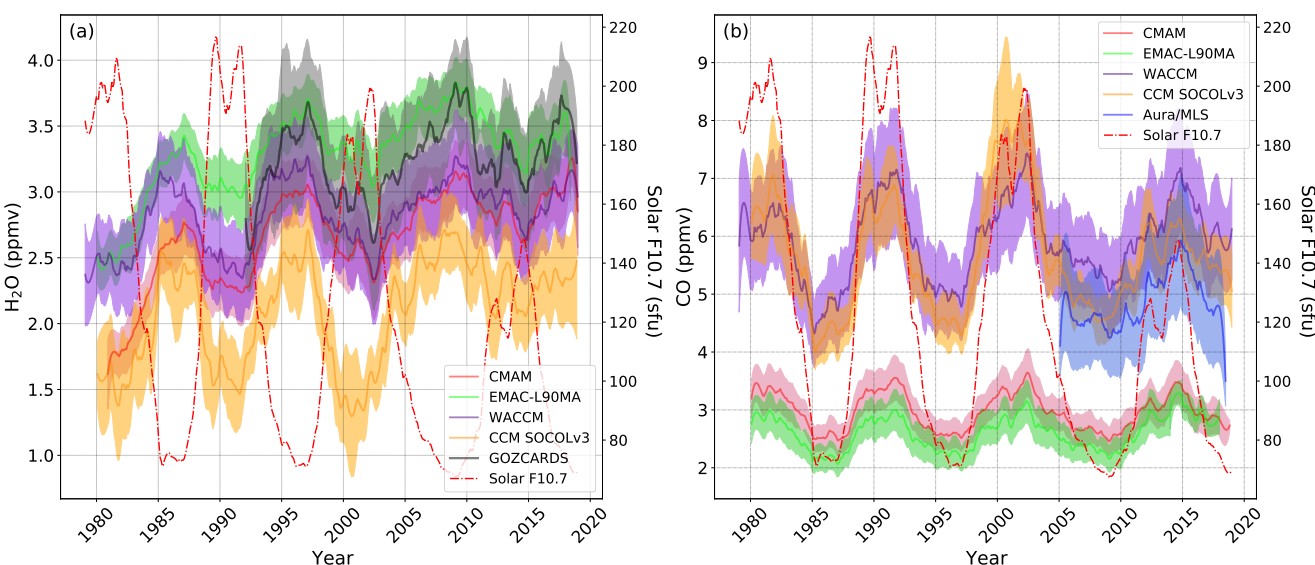

**Figure 1.** Time series of monthly mean (a) $H_2O$ and (b) CO mixing ratio from CCMI-1 models as well as GOZCARDS observational composite (grey line and shading in (a), starting in 1992) and Aura/MLS observations (blue line and shading in (b), starting in 2005) at 0.01 hPa averaged over the tropics ($30^o$N-$30^o$S). Shadings: 1 $\sigma$ standard deviation. The red dash-dotted line indicates the F10.7 solar index.

It should be mentioned that the upper boundary for SOCOL and EMAC at 0.01 hPa belongs to the sponge layer where high diffusion is used to avoid excessive wave amplitudes. The importance for chemistry is that a zero-flux condition is applied for SOCOL and EMAC, which means that $H_2O$ and CO concentrations are not prescribed at 0.01 hPa level. For WACCM

and CMAM, the model top-level is above 0.01 hPa (at $5.1 \times 10^{-6}$ hPa and 0.0008 hPa, respectively) and the influx of the air with rather high CO and low $H_2O$ concentrations from the lower thermosphere could play an important role. For visualization purposes, we smooth $H_2O$ and CO time series presented in Figure 1 using the third-order polynomial interpolation with a 2-year length of the averaging window, however, data used later for MLR analysis are taken in original form without smoothing. As it is shown in Figure 1, there is a pronounced response of $H_2O$ and CO to solar irradiance variability, represented here

as the F10.7 solar radio flux (right vertical axis). In the case of $H_2O$, there is a decrease in mixing ratio during solar activity maximum, and opposite for CO, which is enhanced during the solar maximum. Obviously, the amplitude of the solar signal in $H_2O$ and CO and their mean values are not the same in different models and observations. The comparison of $H_2O$ mixing ratios in Figure 1 during the 1984-2017 period reveals that all models except SOCOL are within the standard deviation of the merged observational data. The observed $H_2O$ mixing ratio is slightly overestimated by EMAC and underestimated by CMAM

and WACCM. A substantial overestimation of the water vapor loss by photolysis in SOCOL may lead to an underestimation of the mixing ratio by up to 50% (Sukhodolov et al., 2017). This can have implications for the simulations of $HO_x$ and ozone loss in the mesosphere. In the case of CO, SOCOL and WACCM results are almost identical and in good correspondence with Aura/MLS observations during 2005-2017. This agreement suggests that the influx of CO from the thermosphere in





WACCM does not substantially contribute to CO in the tropics. However, in SOCOL the lacking of downward transport from the thermosphere might hypothetically be compensated by erroneous, for instance too strong in-situ production in the upper mesosphere. On the other hand, the absolute values of the CO mixing ratio in EMAC and CMAM are very similar. They are underestimated by a factor of 2 though, in comparison to Aura/MLS data, which might be due to an underestimated production. Thus, it is obvious that $H_2O$ and CO behave differently in models and observations, subject of the exact treatment of chemistry and radiation in the models. In the following a detailed MLR analysis of modeled $H_2O$ and CO as well as of the observational data sets will be presented.

## 3 Description of the MLR model

The multiple linear regression (MLR) model used in this study is based on the x-regression tool (Kuchar, 2016) consisting of the Python statistical models library *statsmodels* (Seabold and Perktold, 2010) coupled with the *xarray* package dealing with multi-dimensional arrays (Hoyer and Hamman, 2017). This model configuration adopts a well-established attribution methodology already used in previous studies (Ball et al., 2016; Kuchar et al., 2017). In this version, the MLR model uses 9 explanatory/predictor variables and one response variable which is either $H_2O$ or CO, respectively. As predictors, we use the solar F10.7 index (in solar flux units), the El Niño–Southern Oscillation (ENSO) ERSST v5 Nino4 index (in Kelvin), two Quasi-biennial oscillation (QBO) proxies of at 30 and 50 hPa (in m/s) assuming their orthogonality (Crooks and Gray, 2005), stratospheric aerosol optical depth (SAOD), (dimensionless) as well as two annual (AO) and two semi-annual (SAO) oscillation harmonics. To remove the residual autocorrelation, a second-order autocorrelation (AR2) model is used in an iterative way. The time series of the monthly mean response variables Y(t) (in ppmv) reconstructed as a function of time (t) by the MLR model for every single cell (latitude x pressure level) is:

$$Y(t) = \alpha + \beta SOLAR(t) + \gamma ENSO(t) + \delta_1 QBO30(t) + \delta_2 QBO50(t) + \varepsilon SAOD(t) + 2 - \zeta AO(t) + 2 - \eta SAO(t) + \vartheta TREND(t) + e(t). \quad (5)$$

To estimate the statistical significance of the derived regression coefficients to approximate Y(t), we use a t-test with 95% confidential level taking into account the residual autocorrelation. e(t) in the equation 5 means the stochastic noise of the model where AR2 is included. All explanatory variables with monthly resolution were taken from the KNMI Climate Explorer database[2]. In our study, the regression coefficients for the solar proxy ($\beta$) are estimated using the MLR model as a latitude-altitude matrix, and they are used to calculate the solar signal per 100 units of F10.7 as a percentage of the average value for the whole period as $(\beta/\bar{Y}(t))x100$, where $\bar{Y}(t)$ is an averaged Y for the whole period of interest (in ppmv). As such we estimate the percentage change in $H_2O$ and CO induced by solar irradiance changes from the minimum to the maximum of the 11-year solar cycle. To check how much of the total variability is represented by the solar variability and whether our choice of regressors is justified, we calculate the relative importance (RI) of each regressor. We use the Lindeman-Merenda-Gold measure (LMG, Lindeman et al., 1980) to decompose $R^2$ (coefficient of determination) and to determine RI, which refers to the proportionate contribution each predictor variable makes to the total predicted criterion variance. Figure 2 shows RI

---

[2]KNMI Climate Explorer database, generously made available freely under https://climexp.knmi.nl





distributions of zonally averaged time-series of CO and $H_2O$ at 0.01 hPa between 30°S and 30°N for the period 2005-2017
and 1992–2017, respectively. Our MRL model, including annual and semiannual harmonics, is able to assess 70-90% of total
variability (shown as "total" on the right hand side of both panels in Figure 2). The solar variability represents around 10%
of the total variance and it is the strongest after the SAO (~50%) driver of CO and $H_2O$ variability in all model data and
observations around the equator at 0.01 hPa. While the solar RI in the CO time-series of EMAC and SOCOL agrees well with
the Aura/MLS observations, CMAM overestimates and WACCM underestimates the solar variability. It is worth saying that in
some models AO and SAO in the upper mesosphere may experience some issues, as much of the variability on those timescales
comes from the residual circulation that would not be fully resolved. In terms of the solar RI in the $H_2O$ time-series, EMAC
agrees well with the GOZCARDS dataset. SOCOL together with CMAM overestimates and WACCM rather underestimates
the solar variability. Even larger model spread is revealed in terms of SAO. A significant amount of the SAO variance, much
larger than for the AO at 0.01 hPa, is consistent with a general understanding of the mesospheric variability (Baldwin et al.,
2001). This may be related to the gravity wave drag imposed in the mesosphere and/or its damping (Rind et al., 2014), or the
mesospheric QBO (MQBO) is not as robust as SAO in the mesospheric region as previously thought (Pramitha et al., 2019).
The SAO dominance at 0.01 hPa cautions us against using deseasonalizing methods only with annual cycle (Deng and Fu,
2019). Therefore, in this study we exclude the deseasonalization procedure from the MLR set-up. Only in this way, our model
is able to assess 70-90% of total variability.





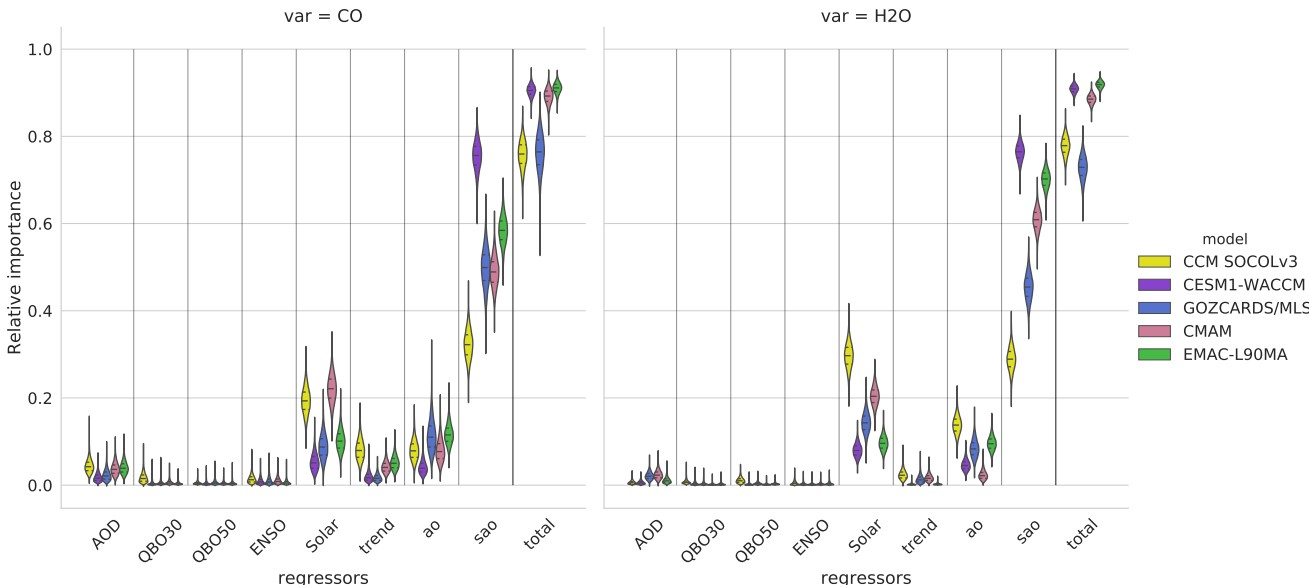

**Figure 2.** The full decomposition of $R^2$ from MLR of equatorial (30°N-30°S) CO for the period 2005–2017 and $H_2O$ for the period 1992–2017 at 0.01 hPa in a form of violin plots. For CO observations the Aura/MLS data are used, for $H_2O$ the GOZCARDS composite. Distributions were calculated from 10000 bootstrapped samples using the LMG measure. Horizontal dashed lines represent quartiles of the distributions. Note that to quantify relative importance of the annual (AO) and semiannual (SAO) oscillation, we do not use deseasonalized time-series.

## 4  Results

### 4.1  Simulated $H_2O$ and CO responses to solar irradiance variability for the 1984-2017 period

Results of the MLR analysis of the $H_2O$ time series from the four CCMs under consideration are presented in Figure 3 for the full investigated time, 1984-2017, while comparisons with observations are shown in Figure 5 for a restricted period.



**Figure 3.** The relative importance of the solar signal in $H_2O$ from CCMI-1 models (1984-2017) presented as percentage of the mean. Model names are indicated at top of each panel. Inclined hatches: area with statistical significance less than 95%.

The most pronounced effect in $H_2O$ is seen in SOCOL and WACCM over the $30^oN$-$30^oS$ latitude band, which appears in the most sunlit region. The effect in SOCOL exceeds those from any other models, with up to a 45% $H_2O$ response to solar irradiance variability. Such a large relative response in SOCOL can be explained by the low background water vapor mixing





ratio (see Figure 1), wider nudging region or by the photolysis by Ly-α implemented in the model that is too intense. The $H_2O$

responses simulated with CMAM and EMAC are smaller and do not exceed 20%. The maximum of the response is slightly

shifted towards the north in CMAM, EMAC and WACCM models as well as the second maximum in SOCOL, which may be

connected to an enhanced residual circulation modulated by the solar cycle (Cullens et al., 2016). The increased downward

propagation of the solar signal can also be found in the WACCM results, where the maximum is also a bit displaced to the

north along with a strengthened descending motion over the north pole. The response of $H_2O$ to solar irradiance variability

disappears below 0.1 hPa in all models because solar irradiance of the Ly-α line cannot penetrate to this depth in the atmosphere

and the influence of the Schumman-Runge band is less substantial.



**Figure 4.** The relative importance of the solar signal in CO from CCMI-1 models (1984-2017) presented as percentage of the mean. Model names are indicated at top of each panel. Inclined hatches: area with statistical significance less than 95%.

Figure 4 shows the estimated CO response to solar irradiance variability in the models for the full period 1984-2017. The similar behavior in CMAM, EMAC, and WACCM suggests a decent of air enriched in CO and a large correlation with solar irradiance over the high latitudes. The penetration is deeper over the southern hemisphere where a stronger southern polar





vortex provides more intensive downward motion and stronger isolation from the middle latitudes. A stronger meridional
transport induced by enhanced atmospheric wave-breaking appears to suggest a maximum CO response over middle and high
latitudes in the northern upper mesosphere (Cullens et al., 2016; Lee et al., 2018). In contrast, SOCOL generates three maxima
of CO (40°S, 40°N and 80–90°N) in the upper mesosphere between 0.01-0.1 hPa, which are not seen in the other models.
Below we will see that this feature depends on the exact time period chosen for comparison (see Figure 7 below). SOCOL
also shows two regions at southern and northern midlatitudes with stronger response and statistical significance above 95%.
Again, the exact appearance of this feature depends on the exact years chosen for averaging (see Figure 7 below). In SOCOL a
sharp boundary in the CO response is seen between 0.1-0.2 hPa due to the lower lifetime of CO there (the OH concentration is
higher) that is too short to allow mesospheric CO to be transported down. This effect can be found in the other models as well,
but only in the 40°S-40°N latitude band. The shape of the solar signal in CO is characterized by a much deeper propagation
over the middle and high latitudes, and it substantially differs from the solar signal in $H_2O$, which is mostly confined to the area
above 0.1 hPa exposed to solar UV in the Ly-$\alpha$ line (dissociating $H_2O$ according to Reaction 2). The reason for the difference
in patterns of $H_2O$ and CO could be a longer chemical lifetime of CO produced by Ly-$\alpha$ in the mesosphere over middle and
high latitudes that allows for transport down through atmospheric circulation.

### 4.2 Simulated and observed $H_2O$ and CO responses to solar irradiance variability

To evaluate the model performance, the simulated solar signals in $H_2O$ and CO are compared with satellite measurements. As
the observations are not available for the full-time period described in the previous sections, we repeated the MLR calculations
using the GOZCARDS merged $H_2O$ data for the 1992-2017 and MLS CO time series for the 2005-2017 periods. The solar
signals in $H_2O$ extracted from the slightly shorter period are illustrated in Figure 5. For none of the models do the simulated
results depend strongly on the time period. The solar signal in $H_2O$ extracted from the satellite data does not show a strong
equatorial response in $H_2O$, as it is visible in most of the model results. Instead, more pronounced effects are shifted to mid-
latitude zones where strong downwelling propagates the solar cycle signal to lower levels. The effects are very similar to those
presented by Remsberg et al. (2018), who also obtained maximum responses shifted to the middle latitudes. The reason of
such a pattern in UARS/HALOE could be related to the sampling issue over the low-tropical region. The same but with a less
pronounced shape appears in the SOCOL results. In this case, the southern maximum is shifted to approximately 20°S and
the northern maximum is shifted to high latitudes in the northern hemisphere. Nevertheless, in terms of percentage, WACCM,
CMAM and EMAC $H_2O$ results are closest-to-observations over the tropical zone. However, over the middle latitudes, only
SOCOL shows a slight poleward shift of the maximum $H_2O$ response, similar to but not quite the same as in the observations,
possibly resulting from the full-atmosphere nudging applied in SOCOL).





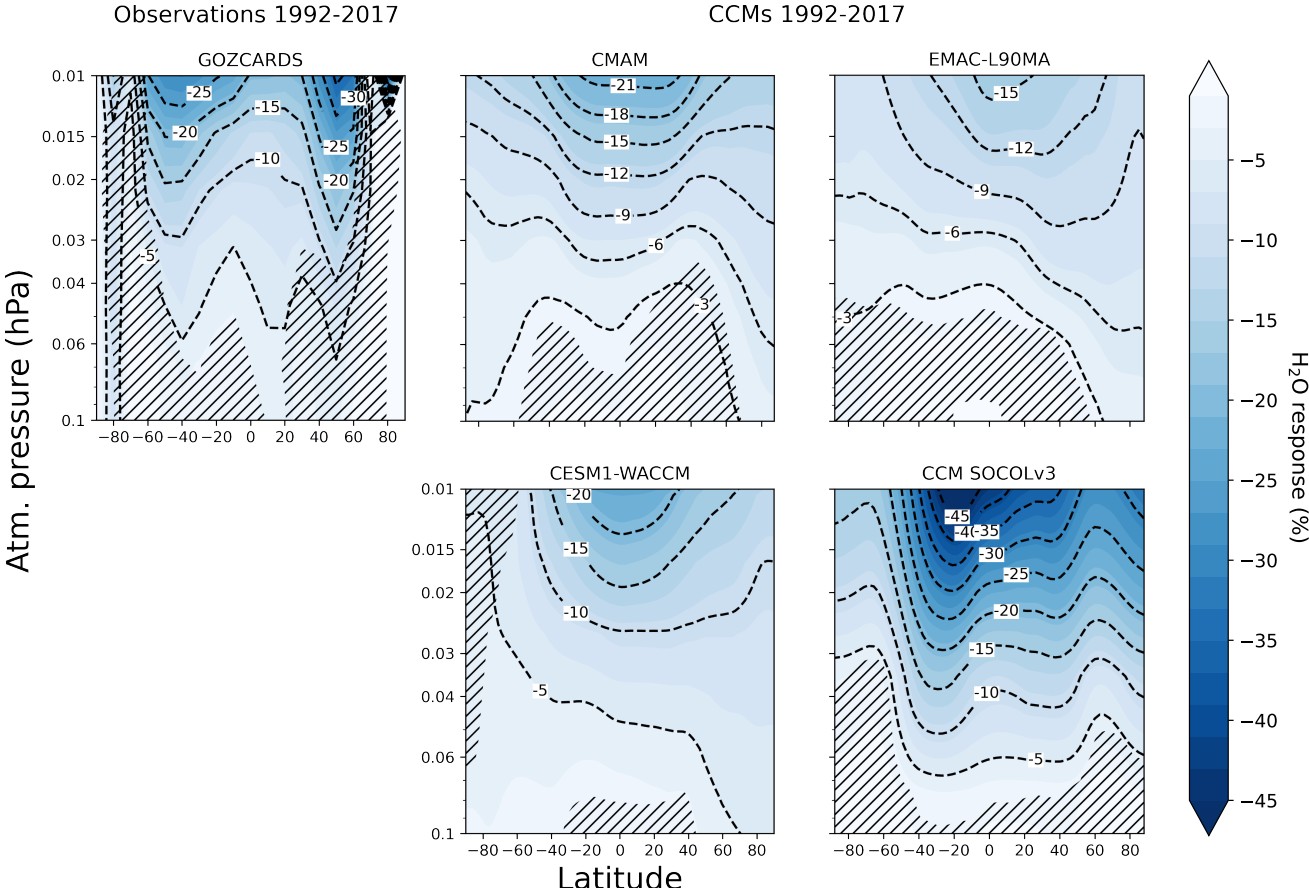

**Figure 5.** The relative importance of the solar signal in $H_2O$ from CCMI-1 models and observations collected by GOZCARDS for the period 1992-2017 presented as percentage of the mean. Model and observations names are indicated at top of each panel. Inclined hatches: area with statistical significance less than 95%.

Because the latitudinal distribution can be related to the peculiarities of the satellite observations such as gaps and measurement inaccuracies, the tropical averaged plot could be more instructive for the evaluation of the model performance. Figure 6 shows the tropical response in $H_2O$ as percentage of the mean, and the change in mixing ratio (in ppmv) averaged over $30°$S–$30°$N. This is the variation of solar irradiance is the largest, and it is less sensitive to thermospheric processes since there is no downwelling over the tropics. To make a better comparison of model results and observations, we present them not only as a ratio to the mean but also as absolute values of solar regression coefficients, since the background water vapor concentrations in the considered datasets are different.





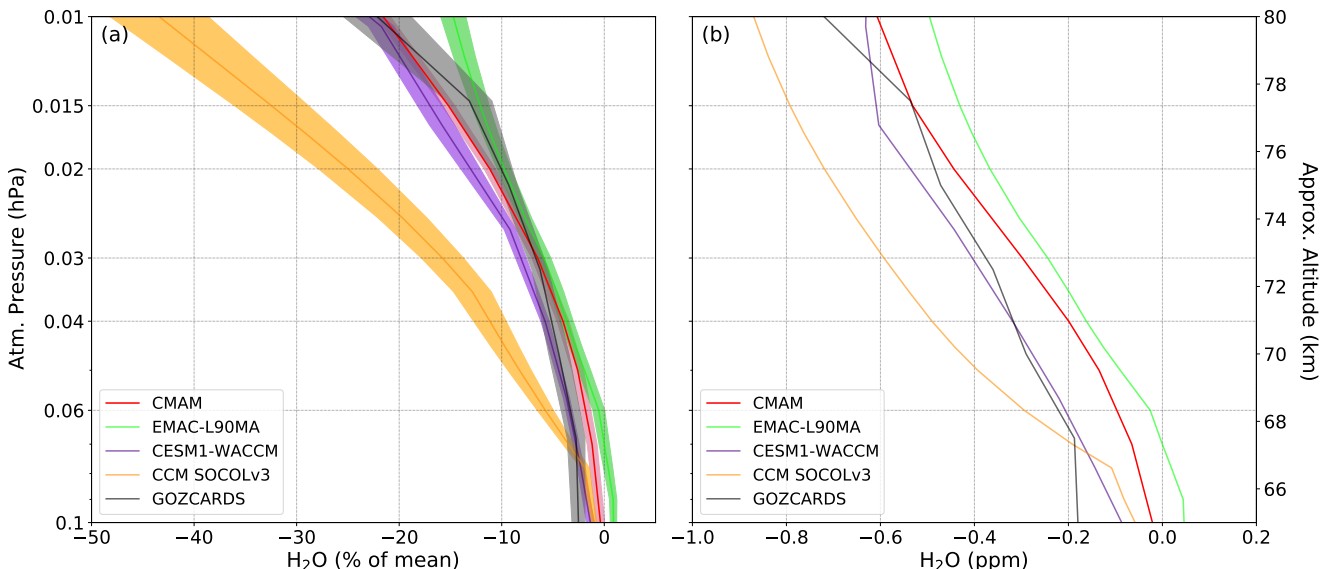

**Figure 6.** Vertical profiles of solar irradiance response in $H_2O$ from CCMI-1 models and GOZCARDS observational composite for 1992-2017 at tropical latitudes (30°N-30°S). (a) The relative importance of the solar signal in $H_2O$ presented as percentage of the mean; (b) $H_2O$ regression coefficient at the solar proxy (β) in mixing ratio (ppmv). Shadings: standard deviation.

In the tropics, the observations show a steady increase of the $H_2O$ sensitivity to the solar irradiance from 0.1 to 0.01 hPa where it reaches the maximum for both, a relative (23%) and absolute (0.75 ppmv) values. Our results agree rather well with the results presented by Remsberg et al. (2018) and Nath et al. (2018). The simulated relative sensitivity values agree well with the observations. However, the SOCOL model shows a much stronger (up to 43%) water vapor sensitivity to solar irradiance (compared with the observed 23%). EMAC results slightly underestimate the observed values, while WACCM and CMAM

show a slightly larger sensitivity. Almost the same pattern is visible for absolute sensitivity values. CMAM and WACCM show the best agreement, while SOCOL and EMAC sensitivities are too strong or too weak, respectively.





**Figure 7.** The relative importance of the solar signal in CO from CCMI-1 models and Aura/MLS for 2005-2017 period presented as percentage of the mean. Model and observations names are indicated at top of each panel. Inclined hatches: area with statistical significance less than 95%.

The solar signals in CO extracted from from the REF-C1SD simulations and observed by MLS data for the 2005-2017 period are illustrated in Figure 7. Opposite to the $H_2O$ case, the influence of the time interval is substantial. The comparison of the results from Figure 4 and Figure 7 reveals that the southern mesospheric maximum of the CO response to solar irradiance variability in SOCOL disappeared, while the northern one became more pronounced. The downward propagation in SOCOL is also intensified and a large and statistically significant solar signal is visible in the upper and middle stratosphere. In CMAM and EMAC the maximum mesospheric response is shifted from the northern mid-latitudes to the equatorial area. There are two peaks of the signal in EMAC, the stronger one is over the equator, but the second one is similarly shifted as in SOCOL and WACCM, showing a maximum at the same pressure levels (from 0.01 hPa to the bottom of the mesosphere) and placed at the same latitude, but both less intensive than in SOCOL. The downward propagation is visible only over the high northern latitudes in CMAM and almost disappears in EMAC. The shape of the solar signal simulated with WACCM does not change



the location; it has a stronger maximum over the middle latitudes, and downward propagation is only marginally significant. This can either be explained by the shortening of the period that put emphasis on some unexplored change, or by different circulation patterns during the 2005-2017 period. The Aura/MLS data shows a maximum in the equatorial middle mesosphere
and middle stratosphere over the high southern and northern latitudes. In the mesosphere, Aura/MLS data are in a better agreement with CMAM and EMAC, while below 0.1 hPa all models equally resemble Aura/MLS observations. Some similarity of the stratospheric response in all considered models and MLS probably results from the applied nudging and therefore it is dynamically induced, contrary to the mesosphere, where the dynamic is only partly nudged, and the models differ substantially.

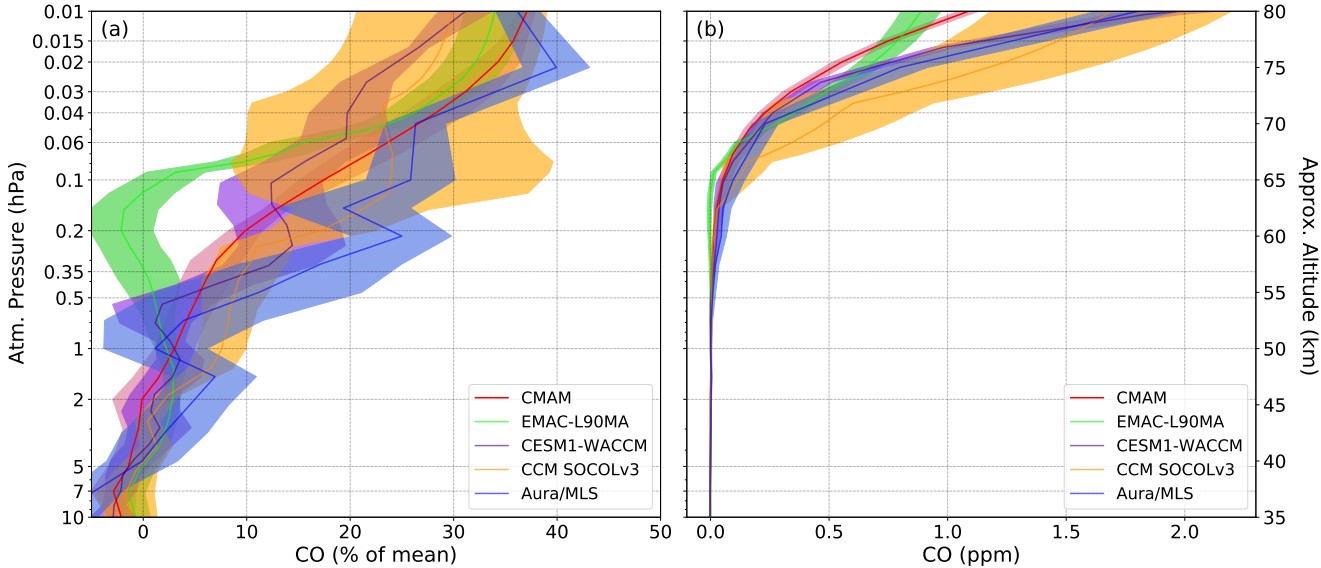

**Figure 8.** Vertical profiles of solar irradiance response in CO from CCMI-1 models and Aura/MLS observations for 2005-2017 at tropical latitudes (30°N-30°S). a) The relative importance of the solar signal in CO presented as percentage of the mean; b) CO regression coefficient at the solar proxy (β) in mixing ratio (ppmv). Shadings: standard deviation.

Figure 8 shows the tropical response in CO as relative change (in percentage of the mean), and the mixing ratio (in ppmv)
averaged over 30°N-30°S. The relative CO sensitivity to the solar irradiance variability averaged over the tropical area from the Aura/MLS data shows a positive correlation from 10 to 0.01 hPa with a magnitude of up to 40% at the mesopause. The simulated sensitivity is within the uncertainty range of the observations for all models except EMAC between 0.35 and 0.06 hPa. The observed absolute sensitivity in the tropical area reaches almost 2 ppmv at the mesopause and is better reproduced by SOCOL and WACCM.



## 5  Discussion

The comparison of absolute values (mixing ratio) of the solar signal in $H_2O$ from models and merged UARS/HALOE and Aura/MLS (GOZCARDS) observations with previous studies reveals higher values in our study for almost all datasets. Comparing the tropical profile plot of $H_2O$ with one from Nath et al. (2018) over the same tropical region (30°S–30°N), it is seen that only EMAC shows a similar magnitude of the solar signal of -0.56 ppm in $H_2O$ from Nath et al. (2018), while all other profiles show stronger responses, including GOZCARDS, which shows -0.73 ppm at 0.01 hPa. However, Nath et al. (2018) used only Ly-$\alpha$ as a solar forcing, yet in the mesosphere other wavelengths contribute significantly to $H_2O$ photolysis and the solar signal in $H_2O$. The latitude-height distribution of the solar signal in $H_2O$ from GOZCARDS and its magnitudes are in a good agreement with Figure 11 of Remsberg et al. (2018), showing similar mesospheric maxima of about 35% over 50°-60°N and a minor maximum of about 25% around 40°S. A comparison with Remsberg et al. (2018) also showed similar features revealed by the MLR setup in our study. Our MLR analysis of Aura/MLS CO shows a weak solar signal of about 40% in the mesospheric CO over the tropics, compared to the solar signal in CO of 68% from Lee et al. (2018). Also, our results show a better representation of the CO solar signal in WACCM for the period 2005-2017 in comparison with the one from Lee et al. (2018). Our results suggest that there is no dominating thermospheric influence of solar irradiance on CO as stated by Lee et al. (2018), because the signal in SOCOL CO shows reasonable results compared to WACCM CO and Aura/MLS observations. However, as it was mentioned above, in SOCOL the absence of thermospheric source of CO could be compensated by overproduction of CO in the upper mesosphere.

However, our MLR analysis revealed a peculiar shift of the solar signal in SOCOL and WACCM, as well as a secondary peak in the same place in EMAC CO for the same period as for Aura/MLS. The nature of this probably reflects the peculiarities of the model dynamics in the Northern hemisphere, which are in some way changed in SOCOL, WACCM, and EMAC during 2005-2017 compared to the longer 1984-2017 period. For the longer period, all models show a similar stronger signal to be shifted northward and downward, and only in SOCOL the solar signal in CO does not reach levels below 0.02 hPa. Among the reasons we suggest, we categorize variations on decadal timescales that may have been attributed as the solar signal, such as global warming, accelerated Brewer-Dobson circulation (BDC), or even changes through sudden stratospheric warmings that facilitate a downward transport of air from the mesosphere. Also, as it was mentioned above, in SOCOL, the nudging is applied for the whole model atmosphere (1000-0.01 hPa) that could make the representation of a dynamical effect on solar signal more reliable. The period of time is also could play a role as 2005-2017 period is rather short for MLR analysis of solar signal since this period is equal to the duration of only one solar cycle. It is important to mention that the signal in Aura/MLS does not show this shift, which makes it more difficult to understand its nature. The latitude-height distributions of the solar signal in $H_2O$ and CO from CMAM, SOCOL, EMAC and WACCM for different periods show that the patterns are very different. The impact of the time period on our results should not be related to the aliasing of regressors, as reported in previous studies (Chiodo et al., 2014; Kuchar et al., 2017), because of the absence of any major volcanic eruptions after 2005.

Our analysis revealed deviations of simulation results from observations showing the weakness of current models in the representation of the solar signal. Our hypothesis is that the major problem is the model dynamics; this issue can be addressed





by the application of more accurate dynamics and transport routines in models. Also, the MLR analysis revealed some incon-

sistencies in the solar signal presented in both, absolute and relative changes compared to observations. For example, SOCOL shows a higher tropical solar signal in $H_2O$ compared to GOZCARDS (Figure 6), but $H_2O$ time series (Figure 1) show lower absolute values by about 2 ppm compared to observations.

One possible reason for the underestimation of $H_2O$ in SOCOL is that only the $H_2O + h\nu \rightarrow H+OH$ photolysis reaction is considered. It is known that $H_2 + O$ products are also possible with about 10% quantum yield, although the much longer

lifetime of $H_2$ should rather lead to less intensive recombination of the products and even smaller $H_2O$ concentration.

However, in the case of CO, SOCOL shows reasonably good absolute values and solar signals in both presented forms compared with the Aura/MLS CO in Figure 8. In the case of EMAC, a weaker solar signal in $H_2O$, despite acceptable absolute values as seen in Figure 1, is simulated. CMAM and EMAC CO show smaller absolute values as presented in Figure 1, and weak solar signals in CO in both, absolute values and % of the mean view, as shown in Figure 8. WACCM simulates a lower

absolute value of $H_2O$ and higher CO compared to the observations, showing a higher solar signal in CO and a lower solar signal in $H_2O$ at 0.01 hPa, correspondingly. Our results show that the transport of CO from the thermosphere, where CO is formed by EUV/soft X-rays photodissociation of $CO_2$, is not much of importance; this is seen by comparing absolute values of CO and the results of our MLR analysis between SOCOL and WACCM, in which thermospheric sources of CO are included. Surely, this is fair to say only for the periods considered here and for the used MLR set-up. $CO_2$ is photolyzed by the Shuman-

Runge continuum (SRC) too, but for SOCOL and EMAC, it does not have an impact since SRC plays a role in the thermosphere that is neither included in SOCOL nor in EMAC, which both have an upper model border at 80 km.

Any impact of volcanic activity upon CO in the upper mesosphere is not likely. However, the large eruptions that occurred around solar maxima, e.g. El Chichón in 1982 and Mt. Pinatubo in 1991, could have some minor effect on the solar signal in CO due to the aliasing effects (Chiodo et al., 2014; Kuchar et al., 2017), however this should not be a problem after 1996.

As such, these issues inspire moderate corrections to model radiation and chemical modules, but which corrections are needed strongly depends on each model, as evidenced by our MLR analysis. We assume that an in-depth comparison of these modules will be needed to find all differences between the CCM set-ups. It might be an option to combine the different approaches of simulation the solar signal in one selected model for further analyses. Moreover, it is needed to use the MLR analysis (or more advanced methods of regression analysis) to check the results of simulations from this MLR analysis. The

comparison of these results between themselves and with available observations could help much to identify the potential ways for model corrections. Also, as the way to reveal problems, especially in dynamic, the comparison of the solar signal from observations can be undertaken with not only model simulations in SD mode but also with free-running model simulations.

## 6 Summary

Using an MLR model, this work extracted and investigated the solar signal in the time series of monthly averaged mixing ratio

of $H_2O$ and CO from CMAM, EMAC-L90MA, SOCOLv3, and CESM1-WACCM 3.5 REF-C1SD model simulations as well as from UARS/HALOE and Aura/MLS measurements. The solar signal was obtained for three periods: for the 1984-2017 period





to compare models between themselves, for the 1992-2017 period to compare the solar signal in $H_2O$ from models against one from merged UARS/HALOE and Aura/MLS (GOZCARDS) observations, and for 2005-2017 to compare the solar signal in CO from models against one from Aura/MLS. As expected, the results of our analysis show that the intensity of the signal increases upward throughout the mesosphere with a maximum at 0.01 hPa in model data and observations of $H_2O$ and CO. However, as our analysis is limited to 0.01 hPa, the actual maximum could be higher. Thus, the variability of $H_2O$ and CO in the mesosphere is strongly determined by the solar irradiance variability over the 11-year solar cycle, with a decrease in $H_2O$ and an increase in CO at solar maximum, and vice versa during solar minimum. Also, our results suggest that atmospheric transport is important for the latitudinal distribution of the considered species with a high sensitivity to solar irradiance variability. The comparison of the latitude-pressure distribution of the solar signal in $H_2O$ for the 1992-2017 period between models and observations shows that the SOCOL model demonstrates a good agreement with the signal of the GOZCARDS observations, yet with a different signal strength. In the case of CO for 2005-2017, the better representation is given by the CMAM model since WACCM and SOCOL show an unexpected shift of the signal to the North. The solar signal in EMAC CO is close to Aura/MLS, but has a second peak in the same latitude range as WACCM and SOCOL. The line plots over the tropics in Figures 6 and 8, both in absolute and relative terms, show similar model results compared to observations in CMAM and WACCM $H_2O$ as well as in SOCOL and WACCM CO.

Overall, our analysis of the solar signal in $H_2O$ and CO shows that the solar signal response in the tropics is confined to the mesosphere as we analyzed the solar signal up to 80 km. The $H_2O$ and CO solar signals over the tropics decay with decreasing altitude and become negligible close to the stratopause in all considered data sets. Besides 10% of the variance attributed to the solar signal variability, the semiannual oscillation dominates the tropical mesosphere.

To sum up, our study demonstrates how state-of-the-art models represent solar signal responses, but also what the weak points of model simulations are. The inter-comparison showed limitations in current simulations, which require a process-oriented validation involving the model teams. These findings strongly suggest to continue the model inter-comparison studies as those within SPARC, IGAC and SOLARIS-HEPPA to improve the representation of the solar signal in global CCMs.

**Data availability**

We provide all LMG results on the Mendeley Data portal (Kuchar, 2020). The CCM results are generally available at the CCMI-1 data archive (http://data.ceda.ac.uk/), except for CMAM, which is available here: ftp://crd-data-donnees-rdc.ec.gc.ca/pub/CCCMA/dplummer/CMAM39-SD_month/. The SOCOLv3-SD data used in this study are not available at the British Atmospheric Data Centre (BADC), they are in general not publicly available. At the BADC, CCMI-1 SOCOLv3 data can only be found.

*Acknowledgements.* AK-D, ER, and WB acknowledge support from the Swiss National Science Foundation under grant 200020-182239 (POLE). AK acknowledges support from Deutsche Forschungsgemeinschaft under grant JA836/43-1 (VACILT). MK acknowledges sup-



port by the Deutsche Forschungsgemeinschaft (DFG) through grant KU 3632/2-1. EMAC-L90MA simulations have been performed at the German Climate Computing Centre (DKRZ) through support from the Bundesministerium für Bildung und Forschung (BMBF). SOCOLv3

simulations were performed on ETH's Linux cluster Euler, partially supported by C2SM grant. DKRZ and its scientific steering committee are gratefully acknowledged for providing the HPC and data archiving resources for this consortial project ESCiMo (Earth System Chemistry integrated Modelling). This material is based in part upon work supported by the National Center for Atmospheric Research, which is a major facility sponsored by the National Science Foundation under Cooperative Agreement No. 1852977. Also, authors, thanks to the SPARC SOLARIS-HEPPA project for the possibility to present and discuss the results of this work during the meeting held during 18-19.09.2019 at

Instituto de Astrofìsica de Andalucía in Granada, Spain.



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
