# Peer review of "The response of mesospheric H2O and CO to solar irradiance variability in models and observations"

_Atmospheric Chemistry and Physics, 2020_

## Referee Comment (RC1) · Anonymous Referee #1 · 22 Sep 2020

This paper is devoted to the connection of water vapor and CO concentration and solar irradiance in the mesosphere in the selected model. The results are presented well, clearly and all new findings the authors try to explain in a good manner. I am not a native speaker but the English in the paper is good. I have not any difficulties with the reading of the paper. and thus according to my opinion this paper can be accepted as is

---

## Referee Comment (RC2) · Anonymous Referee #2 · 5 Oct 2020

The paper is about solar signal in H2O and CO data modelled with CMAM, EMAC-L90MA, SOCOLv3, CESM1-WACCM 3.5 and measured by Aura/MLS and GOZCARDS. The authors extracted the signal from the modelled and measured data using MLR analysis and compared the obtained solar components. In my opinion, the obtained results are interesting and the paper should be published.

Specific comments:

It is not written in the abstract which measurements were used.

lines 123-124. It is written that "For Equation 3, cross-sections from 0.5 to 1050.0 nm in the XUV/X-ray wavelength region are used." It is probably should be 105 nm (not

[Figure]

1050 nm)?

Figure 2. It is written that "While the solar RI in the CO time-series of EMAC and SO-COL agrees well with the Aura/MLS observations, CMAM overestimates and WACCM underestimates the solar variability. " I see that the WACCM is in good agreement with the MLS data. But the data calculated with SOCOL and CMAM are not significantly different. Thus I would say that SOCOL rather overestimates the measurements.

Figure 4. The response calculated with CMAM, EMAC and WACCM for high latitudes lower than 0.2 hPa is large and significant. Why the data calculated with SOCOL do not show any significant response?

Figure 6. I would say that agreement between modelled and measured data depend on the altitude. For example, though EMAC data underestimate the measurements lower than 0.03 hPa they are in perfect agreement at 0.03 - 0.015 hPa. Please, be more specific and make comparison depending on the altitude.

Figure 6. It is written in line 275 that "Almost the same pattern is visible for absolute sensitivity values." I do not agree with this statement. For example, at 0.03 hPa the relative values calculated with CMAM are in agreement with measurements while it is not the case for the absolute values. So, in my opinion, Figure 6b should be described separately and compared with Figure 6a.

Figure 8. It is written in lines 296-298 that "The simulated sensitivity is within the uncertainty range of the observations for all models except EMAC between 0.35 and 0.06 hPa." - The data calculated with WACCM are not within the uncertainty range with MLS between 0.1 and 0.01 hPa.

Technical comments:

line 27. H4 - CH4

line 182-183. two Quasi-biennial oscillation (QBO) proxies of at 30 and 50 hPa (in m/s) - zonal winds at 30 and 50 hPa (in m/s) as proxies of Quasi-biennial oscillation (QBO).

lines 191-192. "they are used to calculate the solar signal per 100 units of F10.7 as a percentage of the average value for the whole period as $(\beta/Y\ Ì\check{D}(t)) \times 100$, where $Y\ Ì\check{D}(t)$ is an averaged $Y$ for the whole period of interest (in ppmv)." - The meaning of the sentence is not clear. Please, explain the statement in more clear way.

lines 266-267. "This is the variation of solar irradiance is the largest, and it is less sensitive to thermospheric processes since there is no downwelling over the tropics. " - The effect of solar irradiance variability is largest in the tropics. Moreover, the $H_2O$ response is less sensitive to thermospheric processes since there is no downwelling over the tropics.

---

## Author Comment (AC1) · 1 Nov 2020

On behalf of all co-authors, I thank you for reviewing our manuscript and for your positive assessment of our paper.

---

## Author Response (AR1)

This paper is devoted to the connection of water vapor and CO concentration and solar irradiance in the mesosphere in the selected model. The results are presented well, clearly and all new findings the authors try to explain in a good manner. I am not a native speaker but the English in the paper is good. I have not any difficulties with the reading of the paper. and thus according to my opinion this paper can be accepted as is

[Figure]

On behalf of all co-authors, I thank you for reviewing our manuscript and for your positive assessment of our paper.

[Figure]

The paper is about solar signal in H2O and CO data modelled with CMAM, EMAC-L90MA, SOCOLv3, CESM1-WACCM 3.5 and measured by Aura/MLS and GOZ-CARDS. The authors extracted the signal from the modelled and measured data using MLR analysis and compared the obtained solar components. In my opinion, the obtained results are interesting and the paper should be published.

Specific comments:

It is not written in the abstract which measurements were used.

lines 123-124. It is written that "For Equation 3, cross-sections from 0.5 to 1050.0 nm in the XUV/X-ray wavelength region are used." It is probably should be 105 nm (not

1050 nm)?

Figure 2. It is written that "While the solar RI in the CO time-series of EMAC and SOCOL agrees well with the Aura/MLS observations, CMAM overestimates and WACCM underestimates the solar variability. " I see that the WACCM is in good agreement with the MLS data. But the data calculated with SOCOL and CMAM are not significantly different. Thus I would say that SOCOL rather overestimates the measurements.

Figure 4. The response calculated with CMAM, EMAC and WACCM for high latitudes lower than 0.2 hPa is large and significant. Why the data calculated with SOCOL do not show any significant response?

Figure 6. I would say that agreement between modelled and measured data depend on the altitude. For example, though EMAC data underestimate the measurements lower than 0.03 hPa they are in perfect agreement at 0.03 - 0.015 hPa. Please, be more specific and make comparison depending on the altitude.

Figure 6. It is written in line 275 that "Almost the same pattern is visible for absolute sensitivity values." I do not agree with this statement. For example, at 0.03 hPa the relative values calculated with CMAM are in agreement with measurements while it is not the case for the absolute values. So, in my opinion, Figure 6b should be described separately and compared with Figure 6a.

Figure 8. It is written in lines 296-298 that "The simulated sensitivity is within the uncertainty range of the observations for all models except EMAC between 0.35 and 0.06 hPa." - The data calculated with WACCM are not within the uncertainty range with MLS between 0.1 and 0.01 hPa.

Technical comments:

line 27. H4 - CH4

line 182-183. two Quasi-biennial oscillation (QBO) proxies of at 30 and 50 hPa (in m/s) - zonal winds at 30 and 50 hPa (in m/s) as proxies of Quasi-biennial oscillation (QBO).

lines 191-192. "they are used to calculate the solar signal per 100 units of F10.7 as a percentage of the average value for the whole period as $(\beta/Y \bar{D}(t)) \times 100$, where Y $\bar{D}(t)$ is an averaged Y for the whole period of interest (in ppmv)." - The meaning of the sentence is not clear. Please, explain the statement in more clear way.

lines 266-267. "This is the variation of solar irradiance is the largest, and it is less sensitive to thermospheric processes since there is no downwelling over the tropics. " - The effect of solar irradiance variability is largest in the tropics. Moreover, the $H_2O$ response is less sensitive to thermospheric processes since there is no downwelling over the tropics.
* * *
Atmos. Chem. Phys. Discuss.,
https://doi.org/10.5194/acp-2020-793-AC2, 2020

[Figure]

Thank you so much for your positive assessment of our paper as well as for useful comments and suggested corrections to our manuscript. We took into account all your comments. Below you can find responses to each of your comments.

QA

Specific comments:

It is not written in the abstract which measurements were used.

Answer: the information about used measurements is added to the abstract.

lines 123-124. It is written that "For Equation 3, cross-sections from 0.5 to 1050.0 nm in the XUV/X-ray wavelength region are used." It is probably should be 105 nm (not 1050 nm)?

Answer: yes, you right, it was a miswrote. The border of the range should be 105 nm. It was corrected in the text.

Figure 2. It is written that "While the solar RI in the CO time-series of EMAC and SO-COL agrees well with the Aura/MLS observations, CMAM overestimates and WACCM underestimates the solar variability. " I see that the WACCM is in good agreement with the MLS data. But the data calculated with SOCOL and CMAM are not significantly different. Thus I would say that SOCOL rather overestimates the measurements.

Answer: yes, it is true. It was corrected in the text.

Figure 4. The response calculated with CMAM, EMAC and WACCM for high latitudes lower than 0.2 hPa is large and significant. Why the data calculated with SOCOL do not show any significant response?

Answer: Among the possible reasons why we do not see this down-propagated solar signal in CO over high latitudes in SOCOL as it is seen in other models could be: only in SOCOL-SD CCMI, the nudging was applied for the whole atmosphere (ERA-Interim data used in SOCOL for nudging are unreliable above 50 km), the vertical resolution (in SOCOL it is 39 levels but, for example, in EMAC it is 90 levels) and the upper boundary that is in SOCOL resides at 80 km.

Figure 6. I would say that agreement between modelled and measured data depend on the altitude. For example, though EMAC data underestimate the measurements lower than 0.03 hPa they are in perfect agreement at 0.03 - 0.015 hPa. Please, be more specific and make comparison depending on the altitude.

Figure 6. It is written in line 275 that "Almost the same pattern is visible for absolute

sensitivity values." I do not agree with this statement. For example, at 0.03 hPa the relative values calculated with CMAM are in agreement with measurements while it is not the case for the absolute values. So, in my opinion, Figure 6b should be described separately and compared with Figure 6a.

Answer: Here the answer is for both the above-mentioned comments. We added a more detailed description of both 6a and 6b figures to the text:

"In EMAC, the relative values of the solar signal in H2O over tropics are well agreed with observations between 0.03 -0.015 hPa but the deviation with the solar signal in observations becomes noticeable above where EMAC underestimates observations. For absolute values, EMAC underestimates solar signal in H2O for the whole presented area. Contrary to EMAC, SOCOL overestimates solar signal in H2O similarly for both relative and absolute values after about 0.05 hPa but underestimates it below 0.03 hPa. In WACCM, the solar signal in H2O mostly located within the observational uncertainty but for relative signal WACCM shows the pronounced deviation from observations within 0.04 and 0.013 hPa, correspondingly. For 0.01 hPa, the WACCM and CMAM correspond well with observations in both relative and absolute value of the solar signal, however showing underestimation in absolute value within the observational uncertainty, though. In the case of absolute values, CMAM agrees well with observations above and underestimates them below 0.02 hPa but in the relative meaning, CMAM H2O underestimates observed solar signal below 0.04 hPa and shows overestimation between 0.25 and 0.015 hPa, respectively."

Technical comments:

line 27. H4 - CH4

line 182-183. two Quasi-biennial oscillation (QBO) proxies of at 30 and 50 hPa (in m/s) - zonal winds at 30 and 50 hPa (in m/s) as proxies of Quasi-biennial oscillation (QBO).

lines 191-192. "they are used to calculate the solar signal per 100 units of F10.7 as a

percentage of the average value for the whole period as ($\beta$/Y ÌD(t))x100, where Y ËǦ ÌD(t) is an averaged Y for the whole period of interest (in ppmv)." - The meaning of the sentence is not clear. Please, explain the statement in more clear way.

Answer: ...they are used to calculate the solar signal as ($Y_s$/ Ym )x100%, where Ym is an averaged $H_2O$/CO (ppmv) for the whole considered period and $Y_s = \beta$ *100 is $H_2O$/CO change (ppmv) caused by F10.7 change by 100 units.

lines 266-267. "This is the variation of solar irradiance is the largest, and it is less sensitive to thermospheric processes since there is no downwelling over the tropics. " - The effect of solar irradiance variability is largest in the tropics. Moreover, the H2O response is less sensitive to thermospheric processes since there is no downwelling over the tropics.

Answer: All your technical suggestions are taken into account and incorporated into the text in the same manner as they were suggested.

Summary
10/11/2020 12:43:58

Differences exist between documents.

**New Document:**
acp-2020-793-manuscript-revised-version
27 pages (5.53 MB)
10/11/2020 12:43:53
Used to display results.

**Old Document:**
acp-2020-793-manuscript-version2
27 pages (5.53 MB)
10/11/2020 12:43:53

Get started: first change is on page 1.

No pages were deleted

**How to read this report**

**Highlight** indicates a change.
 indicates deleted content.
▲ indicates pages were changed.
↔ indicates pages were moved.

[revised manuscript text omitted]